# Hardware-aware Exponential Approximation for Deep Neural Networks

**Xue Geng**[§]**, Jie Lin**[§]**, Bin Zhao**[♮]**, Zhe Wang**[§]**, Mohamed M. Sabry Aly**[♯]**, Vijay Chandrasekhar**[§]
[§]I²R, A⋆STAR, Singapore    {geng_xue,lin-j,wang_zhe,vijay}@i2r.a-star.edu.sg
[♮]IME, A⋆STAR, Singapore    zhaobin@ime.a-star.edu.sg
[♯]School of CSE, NTU, Singapore    msabry@ntu.edu.sg

## Abstract

In this paper, we address the problem of cost-efficient inference for non-linear operations in deep neural networks (DNNs), in particular, the exponential function $e^x$ in softmax layer of DNNs for object detection. The goal is to minimize the hardware cost in terms of energy and area, while maintaining the application accuracy. To this end, we introduce Piecewise Linear Function (PLF) for approximating $e^x$. First, we derive a theoretical upper bound of the number of pieces required for retaining the detection accuracy. Moreover, we constrain PLF to bounded domain in order to minimize bitwidths of the lookup table of pieces, resulting in lower energy and area cost. The non-differentiable bounded PLF layer can be optimized via the straight-through estimator. ASIC synthesis demonstrates that the hardware-oriented softmax costs 4x less energy and area than the direct lookup table of $e^x$, while with comparable performance on benchmark datasets.

## 1 Introduction

Deep neural networks (DNNs) has achieved massive success in a broad of applications in computer vision Krizhevsky et al. (2012), NLP, speech, games Chouard (2016); Silver et al. (2016), etc. This has also triggered rapid development of customized hardware for accelerating the inference phase of DNNs Han et al. (2016); Chen et al. (2017); Jouppi et al. (2017), by hardware-algorithm co-optimization. The seminal work is the custom Application-Specific Integrated Circuit (ASIC) called Tensor Processing Unit (TPU) Jouppi et al. (2017), which is 15x to 30x faster at inference than the general purpose NVIDIA GPU K80. To optimize DNNs algorithm for inference speed up in hardware, recent work mainly focused on reducing either the size of multiply-and-accumulate (MAC) operands (e.g. reduce bitwidth of weights and activations) or the number of MAC operations (e.g. weight or channel pruning), as MAC operations in linear CONV and FC layers account for over 99% of total operations in modern DNNs such as CNN and LSTM.

It is worthy noting that non-linear operations (e.g. softmax and sigmoid) could also be identified as the computational bottleneck of DNNs. An example is the softmax function $\sigma(x_c) = e^{x_c}/\sum_{j=1}^{C} e^{x_j}$, which is widely used in modern DNNs for different applications, e.g. Faster R-CNN for object detection. In the classification stage of Faster R-CNN, the cost of computing softmax is proportional to {# bounding boxes to be evaluated} × {# detection classes}, which easily becomes the key challenge for inference phase of Faster R-CNN (e.g. $300 \times 10,000 = 3$million per image). However, there is very few publicly available research work optimizing non-linear operations for DNNs-specific hardware. Google TPU developed the *activation* unit to support non-linear operations Jouppi et al. (2017), but without revealing technical details.

In this paper, we aim to develop cost-efficient inference mechanism for non-linear operations in DNNs. In particular, we are interested in approximating $e^x$ in softmax layer. The approximation of $e^x$ is expected to minimize the energy and area cost, without incurring considerable performance loss in accuracy. To fulfill all these constraints, we propose a bounded Piecewise Linear Function (PLF) for approximating $e^x$. The bounded PLF is hardware-friendly in the sense that it is multiplication-free and requires small lookup table size, which are favorable properties for DNNs specific hardware. For instance, 32-bit fixed-point addition can be 30x less energy and 25x less area than 32-bit fixed-

point multiplication Dally (2016). To the best of our knowledge, this is the first publicly available research work towards hardware-aware exponential approximation for DNNs.

One may note that we did not consider the classical polynomial approximation approaches for $e^x$, such as Taylor series Nilsson et al. (2014), which would introduce far more multiplication operations than lookup table based approximation schemes, in order to preserve application accuracy.

## 2 APPROXIMATE $e^x$ WITH BOUNDED PIECEWISE LINEAR FUNCTION

**Preliminary**. Towards hardware-aware exponential approximation, the key is to develop cost-effective approximation for $e^x$. A straightforward approach is Piecewise Linear Function (PLF), which typically approximates non-linear curve with a set of line pieces Amin et al. (1997); Namin et al. (2009). In geometry, PLF approximates $e^x$ with $S$ continuous pieces uniformly defined over a finite domain of $x \in [x_l^1, x_r^S]$, each of those pieces is an affine function with slope $\alpha^s$

$$f^s(x) = \alpha^s * (x - x_l^s) + y_l^s = \alpha^s * x + (y_l^s - \alpha^s * x_l^s)$$
$$x \in [x_l^s, x_r^s], \quad y_l^s = e^{x_l^s}, \quad s \in [1, S] \tag{1}$$

At inference phase, a lookup table of pieces is built to decide which piece is chosen for computing the approximate value $f^s(x)$ in Eq 1, given $x$. The cost of computing $f^s(x)$ contains 1 multiply ($\alpha^s * x$) and 1 addition ($\alpha^s * x_l^s + y_l^s$ is pre-computed and stored in lookup table, together with $\alpha^s$).

In Table 1, we evaluate PLF in terms of detection accuracy on 2 benchmark datasets with 2 state-of-the-art object detection networks. Specifically, we train the DNNs with softmax, and replace $e^x$ in softmax layer with PLF at inference phase. We also include another lookup table baseline (*Lookup-$e^x$*) that directly stores the exact value of $e^x$ over uniformly sampled $x$ in $[x_l^1, x_r^S]$. We observe that PLF requires much smaller table size (4x) than Lookup-$e^x$.

Next, we address 3 remaining issues for PLF. First, we analyze how does lookup table size (the number of pieces) affect application accuracy, in particular, in the context of object detection with DNNs. Second, we propose to constrain the domain of $x$ for PLF (termed Bounded PLF) to save bitwidths of lookup table values, resulting in lower energy and area cost. Finally, we further reduce the energy and area cost of Bounded PLF by removing the multiply operation $\alpha^s * x$.

**PLF: Lookup table size?** We provide a theoretical analysis to derive the upper bound of lookup table size $S$ (i.e. the number of pieces in PLF), without reducing application accuracy of object detection. In the classification stage of object detection networks (e.g. Faster R-CNN or R-FCN), the output of softmax is $C$-d confidence scores for each of the $P$ region proposals (a.k.a. ROI bounding boxes). For each class, the region proposals are sorted by their corresponding confidence scores, followed by the subsequent non-maximum suppression (NMS) for bounding box filtering. To maintain application accuracy with PLF, the sufficient condition is that PLF does not change the order of confidence scores on predicted boxes generated by softmax.

We define the adversarial noise Zhou et al. (2017) between a consecutive pair of region proposals sorted by softmax, $\mathbf{r}^*(x_{p,c}) = ||\sigma(x_{p,c}) - \sigma(x_{p+1,c})||^2/2$, which is the minimum noise that would change the order of the pair. Let $\mathbf{r}$ denotes the error between softmax and the approximate value of PLF for the $i^{th}$ bounding box with the $c^{th}$ class,

$$\mathbf{r}(x_{p,c}) = ||\sigma(x_{p,c}) - \frac{f^s(x_{p,c})}{\sum_j^C f^s(x_{p,j})}||, \qquad p \in (0, P-1) \tag{2}$$

To maintain the order of predicted bounding boxes for class $c$, PLF has $P$ pairwise constraints,

$$||\mathbf{r}(x_{p,c})||^2 + ||\mathbf{r}(x_{p+1,c})||^2 \le \mathbf{r}^*(x_{p,c}), \qquad i \in (0, P-1) \tag{3}$$

Given that the domain of $x$ is fixed for the uniform PLF, Eq 3 is determined by the lookup table size $S$. We use Bellman-Ford algorithm Goldberg & Radzik (1993) to solve these inequalities, and derive a theoretical upper bound for the table size $S$, which enables efficient binary search of the practical minimal table size that retains application accuracy.

**Bounded PLF**. The entries in the lookup table are stored as fixed-point number representations, arbitrary domain of $x$ probably leads to higher bitwidths of entries as well as higher energy and area cost. One can see from Eq 1 that the range of $y_l^s$ depends on the domain of $x_l^s$, e.g. to maximize

Table 1: Comparisons of PLF with softmax and lookup table of $e^x$ on 2 object detection benchmark datasets PASCAL VOC2007 Everingham et al. (2010) and MS-COCO 2014 Lin et al. (2014). Following the standard protocol, we report results on VOC2007 test set and MS-COCO minival set in terms of mean Average Precision (mAP). Both Faster-RCNN and R-FCN with various backbone networks (VGG and ResNet) are pre-trained with softmax. At inference phase, we replace $e^x$ in softmax layer with either PLF or Lookup-$e^x$. For both approximation methods, we report the minimum number of table size that maintains comparable detection accuracy with softmax.

| Methods | min. table size | VOC2007 mAP | | COCO mAP@[0.05-0.95] | |
|---|---|---|---|---|---|
| | | Faster R-CNN (VGG16) | R-FCN (ResNet50) | Faster R-CNN (VGG16) | R-FCN (ResNet101) |
| PLF | 32 | 72.06 | 71.69 | 24.10 | 28.10 |
| Lookup-$e^x$ | 128 | 72.55 | 71.64 | 24.10 | 27.40 |
| Softmax | – | 72.52 | 71.76 | 24.20 | 28.20 |

Table 2: ASIC synthesis experiments to evaluate the hardware cost of approximating $e^x$ in terms of bitwidth of lookup table values as well as area and energy cost. We compare the Bounded PLF (BPLF) with Lookup-$e^x$, using the clipped domain of $x$ with or without training. Results are reported on PASCAL VOC2007 test set with Faster R-CNN + VGG16. Both approaches are synthesised using UMC CMOS 65nm Standard cell library. Note that we adopted hard-coded lookup table with logic gates, as using memory to store lookup table is not efficient for the silicon area if table size is less than 1k entries. Again, we report the minimum number of table size that maintains comparable detection accuracy with softmax. Numbers in bracket represent the number of sign bits, integer bits and fractional bits for entries in the lookup table, respectively.

| Methods | | VOC2007 mAP | min. table size | bitwidths | | | *area* ($\mu m^2$) | *energy* (normalized) |
|---|---|---|---|---|---|---|---|---|
| | | | | $\alpha^s$ | $x$ | $e^x$ | | |
| Clip | BPLF | 71.93 | 8 | (0,4,0) | (1,4,2) | (0,20,1) | 277 | 1.8x |
| ($\gamma = 12$) | Lookup-$e^x$ | 72.59 | 128 | – | (1,4,2) | (0,20,1) | 921 | 6.4x |
| Trained Clip | BPLF | 70.58 | 2 | (1,3,0) | (1,3,2) | (0,10,3) | 20 | 1.0x |
| ($\gamma = 6$) | Lookup-$e^x$ | 70.70 | 64 | – | (1,3,2) | (0,10,3) | 38 | 4.5x |

the precision and avoid overflow, the minimum number of integer bits for $e^{12}$ is 17-bit. To minimize bitwidth, a clipped layer is applied at the bottom of PLF to bound the domain of $x \in [-\gamma, \gamma]$

$$h(x) = min(max(x, -\gamma), \gamma) \tag{4}$$

where $\gamma$ is a pre-defined positive threshold. Smaller $\gamma$ results in lower hardware cost, but may incur performance loss in accuracy at inference phase, due to the precision loss of $e^x$. To alleviate this issue, we propose to train the DNNs with the clipped layer combined with the PLF layer, as the replacement of $e^x$ in softmax layer. As Eq 1 and Eq 4 are non-differentiable functions, we use Straight Through Estimator (STE) Bengio et al. (2013) to enable the back-propagation.

**Bounded PLF without Multiply**. Finally, the multiply term $\alpha^s * x$ in Eq 1 can be also transformed to bit shifting operation, by approximating $\alpha^s$ with $2^b$ where $b$ is integer constant. As a result, the energy and area cost of Bounded PLF are further reduced.

Table 2 compares the Bounded PLF (BPLF) with Lookup-$e^x$ in terms of hardware cost. ASIC synthesis experiments using UMC CMOS 65nm Standard cell library are performed to generate the hardware metrics, including area and energy cost. First, we observe that the trained clipped layer ($\gamma$ reduced from 12 to 6) helps reducing the number of minimum table size as well as the bitwidths of lookup table values, with sacrificing slight mAP. Second, BPLF dramatically outperforms Lookup-$e^x$ with much less area and energy cost (e.g. 2x-4x). These results demonstrate the effectiveness of the proposed BPLF in both hardware and software aspects.

## 3 CONCLUSION

Future work includes (1) extending the approximation of $e^x$ to softmax by taking division into account; (2) beyond optimizing algorithm for hardware, exploring hardware-algorithm co-design for softmax approximation; (3) evaluating softmax approximation on other applications like language modeling. The paper idea can be applied to other nonlinear operations, e.g., sigmoid and tanh.

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
