# OpenReview forum: "Hardware-aware Exponential Approximation for Deep Neural Network"
_ICLR.cc/2018/Workshop — Reject_

### Official Review · AnonReviewer3 · 2018-03-02
**Interesting work, but it is not clear which are the adavantages**

**Rating:** 5
**Confidence:** 4

**Review:**

This paper proposes a piecewise linear function (PLF) to quickly approximate the exponential e^x used in the softmax layer of a neural net. In particular, for object detection, softmax is applied to every boundingbox, and therefore it may be important to reduce its computational cost. Compared to an approaximation based on constant values (instead of linear approximation), PLF can reduce the lookup table size by a factor 4, without reducing the detection performance. The table size can be further reduced using a bounded approximation of e^x and by retraining the detector with the approximate exponential.

Pros:
- The paper is well presented

Cons:
- My main critic to the paper is the very small practical effect of the contribution. I have to say that I am not an expert in hardware, but, in my understanding, even in detection, the real cost of softmax is really marginal compared to convolutions. Thus, optimizing a part that takes only a small fraction of the full computation, to me does not change much in terms of speed or hardware cost. The authors do not enter in this discussion, they just mentioned that in R-CNN for the case of classifying 10k classes, (which is not the case of COCO, 80 classes or VOC 20 classes), the softmax will be used 3M times. However, they do not mention, which is the percentage of cost of this operation compared with the rest. I do not have numbers, but, in my understanding it should be below 1% of the total cost.
- The idea of using a PLF for e^x is a very simple contribution.
- It is not clear how much is the advantage of PLF compared with standard approaches of computing e^x. Even if the approximation of e^x with Taylor series is slower, it would have been interesting to see results with that too.



Additional comments:
- What about using a continuos linear approximation of e^x so that you do not need to use the straight through estimator?
- The section about the estimation of the lookup table size is not very clear to me

---

### Official Review · AnonReviewer1 · 2018-03-09
**Insufficient technical depth and breadth of applicability**

**Rating:** 4
**Confidence:** 3

**Review:**

The authors propose approximating the exponential e^x in softmax layers of DNNs for object detection by a bounded piece-wise linear function (BPLF). They show a theoretical upper bound on the number of pieces for a required accuracy. They train DNNs with softmax and replace e^x with BPLF while doing inference. They analyze how lookup table size affects detection accuracy and also save on the bitwidths of lookup table values using BPLFs. They observe 2x- 4x energy savings on detection benchmark datasets.

I do not see enough technical depth in their approach. I am also unsure about the breadth of this line of work. The authors do mention that the same idea can be applied to various non-linear operations. However, the current experiments are not enough to draw any general conclusion about energy/area-savings along with faster inference.

---

### Decision · Program_Chairs · 2018-03-20
**ICLR 2018 Workshop Acceptance Decision**

**Decision:**

Reject

**Comment:**

Based on the reviews, this paper has not been accepted for presentation at the ICLR workshop. However, the conversation and updates can continue to appear here on OpenReview.